# Unlocking the Genetic Secrets of Pancreatic Cancer: *KRAS* Allelic Imbalances in Tumor Evolution

**DOI:** 10.3390/cancers17071226

**Published:** 2025-04-04

**Authors:** Vasiliki Liaki, Blanca Rosas-Perez, Carmen Guerra

**Affiliations:** 1Molecular Oncology Program, Spanish National Cancer Research Center (CNIO), 28029 Madrid, Spain; brosas@cnio.es (B.R.-P.); mcguerra@cnio.es (C.G.); 2Centro de Investigación Biomédica en Red de Cáncer (CIBERONC), Instituto de Salud Carlos III, 28029 Madrid, Spain

**Keywords:** pancreatic ductal adenocarcinoma (PDAC), KRAS, allelic imbalance, wild-type allele, loss of heterozygosity, tumor evolution, genomic rearrangements, metastasis

## Abstract

Pancreatic cancer is one of the most heterogeneous and highly aggressive malignancies. Traditionally, the gradual evolution model of pancreatic tumors has been based on the progressive acquisition of key genetic alterations. However, a complementary model based on rapid genetic evolution has also emerged, according to which driver gene inactivations can occur simultaneously. In both models, an activating mutation in the KRAS oncogene is the main initiating event in acinar pancreatic cells. However, in advanced stages of the disease, additional amplifications and major imbalances in the mutant KRAS allele are often detected. Here, we discuss how KRAS allelic imbalances can arise during tumor evolution. We also analyze the importance of KRAS allelic imbalances in pancreatic cancer biology and their possible therapeutic implications.

## 1. Introduction

Pancreatic cancer is one of the most aggressive malignancies and has become the third leading cause of cancer-related deaths worldwide [1]. With a 5-year survival rate of only 5%, the median survival is less than 6 months [2]. This elevated mortality is mainly attributed to late diagnosis, as locally advanced disease is largely asymptomatic. Unfortunately, this dismal prognosis has remained unchanged over the years, as minimal improvements have been made in the field of treatment [3]. Currently, surgical resection remains the only curative option for early-stage local disease, which corresponds to 20-30% of patients. Nevertheless, the recurrence of pancreatic tumors occurs in up to 85% of these patients [4]. So far, adjuvant treatment with 5-fluorouracil, leucovorin, irinotecan, and oxaliplatin (FOLFIRINOX) has been demonstrated to provide the longest median overall survival (OS) of 54 months in patients with resectable disease [5].

Approximately 70–80% of patients do not benefit from surgery, because at the time of diagnosis they display advanced disease with metastasis. Thus far, cytotoxic chemotherapy is the standard treatment for late-stage patients, with overall survival in the range of weeks to a few months [3]. More specifically, the standard of care consists of modern combinations of highly cytotoxic agents, such as nab-paclitaxel plus gemcitabine or FOLFIRINOX, that only provide modest improvements in OS, in the range of weeks to months [6,7]. Undoubtedly, there is urgent need for novel targeted therapies with reduced toxicities. At present, there are numerous ongoing clinical trials investigating the efficacy of inhibitors against druggable mutations, such as KRAS, PARP, and SHP2 [8,9,10]. Yet, only few targeted therapies or immunotherapeutic approaches have been approved for clinical use [11].

Pancreatic Ductal Adenocarcinoma (PDAC) accounts for more than 90% of all pancreatic tumors. The majority are non-hereditary PDACs and are caused by the accumulation of somatic mutations in oncogenes and tumor suppressor genes. All large-scale genomic studies have confirmed the most common alterations in the KRAS oncogene, and the tumor suppressor genes, CDKN2A, TP53, and SMAD4 [12,13,14,15]. More specifically, activating mutations in KRAS are detected in 95% of all pancreatic tumors [14,15]. CDKN2A, TP53, and SMAD4 are commonly inactivated in 50–80% of pancreatic tumors by methylation or homozygous deletions [16]. Other mutated genes in ~10% of tumors include TGFBR1, ARID1A, GLI3, and TGFBR2 [16,17].

Additionally, other types of genomic changes, such as copy number alterations, that involve high numbers of amplified or lost genes, are also often detected [17,18]. Waddell et al. have separated pancreatic tumors based on the number and location of structural variants. Subtypes designated as locally rearranged or unstable seem to be defined by a large (>200) number of events or structural variants, respectively [13,16]. These genomic rearrangements occur frequently during tumor evolution, explaining the limited heterogeneity observed among the known driver genes between primary and metastatic PDAC [19].

Here, we present a brief overview of the proposed models of PDAC evolution, in correlation with these genomic alterations. We also focus on the main driver oncogene, KRAS, and discuss how different allelic frequencies arise during tumor progression and how they affect tumor biology.

## 2. Tumor Evolution

Two models of tumor evolution have been proposed for the genomic events that occur during PDAC evolution. According to the classical or gradual progression model, precursor lesions accumulate genetic alterations over many years and eventually lead to malignancy [20,21,22]. Alternatively, the more recent model of simultaneous progression proposes that PDAC driver gene activations and inactivations can occur concomitantly through complex chromosomal rearrangements, leading to rapid tumor development and dissemination [23,24].

### 2.1. Gradual Progression Model 

Most PDACs emerge from non-cystic precursor lesions, such as pancreatic intraepithelial neoplasias (PanINs) [25], although a small minority consists of cystic lesions, such as intraductal papillary mucinous neoplasm (IPMN) and mucinous cystic neoplasm (MCN) [26]. PanINs are microscopic, non-invasive epithelial neoplasias confined within pancreatic ducts, generated from acinar cells through a reprogramming process called acinar-to-ductal metaplasia (ADM) [27,28]. In normal pancreas, ADM is a temporary and reversible process. However, persistent ADM can lead to PanIN development and, eventually, PDAC [29]. The transition from ADM to PanINs has been well-documented in both mouse models and humans [25,28,30,31,32]. It is characterized by the induction of ductal cell identity and the suppression of acinar cell characteristics, both at gene expression and morphology levels [33,34]. Interestingly, PanINs are surrounded by unique microenvironment, are markedly different from ADM, and are transcriptionally distinct from ducts [35]. The earliest genetic event that initiates alterations is the oncogenic activation of KRAS that forces acinar cells into a constant ADM state, which can quickly lead to the formation of PanINs [36,37] (Figure 1).

PanIN lesions are divided into three grades of dysplasia, according to the grade of cytological and architectural atypia. PanIN1A and PanIN1B are classified as low grade, although they are subdivided based on the extent of the structural abnormalities (Figure 1). On the other hand, PanIN2 and PanIN3 present high-grade dysplasia, and PanIN3 are considered in situ carcinomas [28,38,39]. Progression from low-grade to high-grade and, eventually, to locally invasive PDAC is caused by the accumulation of different genetic and non-genetic alterations (Figure 1).

The gradual progression model posits that histopathological lesions are driven by the stepwise accumulation of specific known mutations [21]. As mentioned before, the main driver genes of PanINs and PDAC include KRAS, TP53, CDKN2A, and SMAD4 [14,21,40]. KRAS mutations are present in low-grade PanIN, indicating that it is the earliest alteration that drives ADM [41,42]. More importantly, they are also detected in healthy pancreatic tissue, thus confirming its role as the initiating oncogenic event [35]. Mutations in the tumor suppressor CDKN2A, are also often found early in the tumorigenesis process [43]. Further accumulation of inactivating mutations in the tumor suppressor genes TP53 and SMAD4, leads to progression to PDAC (Figure 1) [44,45,46]. The accumulation of genetic and morphological alterations is proposed to occur during one to two decades until invasive carcinoma is developed [44]. However, although it appears to be a gradual process, not all mutations are present in all patients. The inactivation of different tumor suppressors, their type of alteration, and the combination of both, results in wide tumoral heterogeneity. These evolutionary trajectories give rise to a large number of inter- and intra-tumoral differences that define tumor behavior [44,47,48].

### 2.2. Simultaneous Progression Model

Traditionally, pancreatic tumor evolution is considered a gradual process. However, recent studies on the simultaneous progression model propose that genetic alterations can also occur in a short period of time or even simultaneously (Figure 2) [23,24,49,50]. Such alterations include the synchronous inactivation of tumor suppressor genes and are associated with complex genomic rearrangements (Figure 2) [51,52,53]. Therefore, they can induce accelerated tumor progression and set off invasive cancer, as they are commonly detected in advanced stages of pancreatic tumors (Figure 2) [50,53]. Mechanistically, the processes involving such genomic events include mitotic errors, chromothripsis, and polyploidization [13,49,51]. However, even though these molecular events have been previously described, our understanding of their impact on tumor evolution remains incomplete. 

It is also worth mentioning that periods of latency between morphological states can follow a different order to that of the classical gradual progression model (Figure 2). Considering the lesions found in samples from patients and healthy individuals, it seems that there is a relatively long latency period between PanIN1 and -2, but rapid evolution between PanIN2 and PDAC [23]. In this scenario, PanIN3 may contain all the molecular information for this rapid progression, as in some cases they display higher somatic mutation burden [50,54].

In the simultaneous progression model, two key steps are identified: the cancer-initiating event and the cancer-transforming event [49,50,55]. KRAS mutations act as the initial trigger of pancreatic cancer. Using this as the reference point in the concept of the gradual progression model, it is widely considered that metastasis occurs late in the genomic timeline [44,56]. However, metastasis may not always be a late event, but can arise concurrently with primary tumor development. According to the simultaneous model, this is induced by cancer-transforming events (Figure 2). Such events either induce or are necessary for transformation, and endow cells with both invasive and metastatic abilities, resulting in a short latency period between local expansion and metastasis. Indeed, it was demonstrated that PDAC cells with high metastatic capacity can also appear simultaneously or even earlier than the formation of primary tumors [57,58,59]. More specifically, lineage-labelling studies involving genetically engineered mouse models have revealed the early dissemination of invasive circulating pancreatic cells [57]. Notably, these studies align closely with the clinical observations for some patients with tumors of very small sizes, which present advanced disease and very short survival of less than 2 years [23]. 

### 2.3. Are the Two Models Mutually Exclusive or Complementary?

As Theodosius Dobzhansky wrote, “nothing in biology makes sense except in the light of evolution” and molecular oncology is no exception [60]. Neo-Darwinian cancer evolution models typically assume the sequential acquisition of mutations over time [61], such as the classical progression model of PDAC evolution. In regard to this concept, microevolution involves smaller, incremental genetic changes that occur over time, contributing to the gradual adaptation and diversification of tumor cells within the microenvironment [44,61]. Yet, the classical model of accumulating genetic changes under selective pressure does not fully explain the diverse evolutionary trajectories of pancreatic cancer. 

Alternatively, tumor macroevolution in PDAC can be driven by single “catastrophic” events that cause complex genomic rearrangements involving numerous genetic loci [49,50,62]. Macroevolution refers to large-scale evolutionary changes, such as quick and synchronous chromosomal losses and gains, that lead to significant alterations in tumor biology and behavior [61,62,63]. These combined genomic alterations accelerate tumor evolution far more dramatically than any single type of event alone, creating unstable tumors with aggressive behavior [18,63]. According to the simultaneous model, evidence supports that numerous genomic aberrations can occur in short bursts due to chromosomal instability, breakage–fusion–bridge cycles, chromoplexy, and chromothripsis [62].

Despite the impact of macroevolutionary events, clinical and experimental evidence consistently underscores the "four usual genes" as those subject to the most recurrent alterations in pancreatic cancer patients. This is also supported by the low level of difference among the driver genes and low genetic heterogeneity observed between primary tumors and metastases [19]. However, a more realistic scenario posits an interplay between point mutations in the driver genes, associated with gradual progression, accompanied by the large-scale genomic events, described in the simultaneous model (Figure 3). Early mutations, such as the ones in KRAS, could appear as a point consequence, while the later inactivation or activation of tumor suppressor genes or oncogenes, respectively, could be due to rearrangements of chromosomes and structural variants (SVs) (Figure 3) [64]. Alternatively, oncogenic variants of TP53 and CDKN2A in combination with KRAS can themselves cause chromosomal instability and rearrangements [46,65]. Indeed oncogene-induced DNA replication stress is often responsible for the presence of genomic instability [66]. Additional polyploidization and genome doubling can add up, providing fertile ground for the acquisition of further mutations and increased genomic gains of already amplified loci [64,66,67].

Thus, tumor cells may alternate between prolonged phases of relative mutational stability and short periods of rapid evolutionary change [68] (Figure 3). Therefore, the two models discussed above are not mutually exclusive, but rather coexist in a hybrid way among tumor populations, contributing to the heterogeneity and adaptability of the tumors (Figure 3).

## 3. KRAS Allelic Frequencies in PDAC

### 3.1. Tumor Progression and Metastasis

In agreement with the accelerated tumor development process that we have described, several studies have reported complex genomic rearrangements in a significant percentage of human PDACs [49,51,63]. More specifically, Notta et al. and Mueller et al. reported a high frequency of chromothripsis and copy number alteration events in primary tumors that progressed to advanced disease [49,52]. Not surprisingly, in both studies, among the genetic regions with losses and gains were KRAS and the usual tumor suppressor genes [49,52]. Interestingly, increased genetic dosage of oncogenic KRAS was detected in both human and mouse PDACs [51,52]. Different mutant KRAS gene dosage states were found, including focal gains, arm-level gains, and copy-neutral loss of the wild-type allele [52]. Chan-Seng-Yue et al. described that apart from the increase in the copy number of the mutant KRAS, the complete loss of the wild-type KRAS gene is also a common event (Figure 4) [51]. Therefore, KRAS allelic imbalances in PDAC are defined by the loss of the wt allele (loss of heterozygosity, LOH), amplifications of the mutant, or both (Figure 4) [51,52]. The degree of this imbalance in PDAC is considered a continuum and is often categorized in minor and major imbalances (Figure 4) [51].

Interestingly, it was previously shown that ADM and PanINs display some of the typical genetic alterations of PDAC. These include aneuploidy, heterozygous mutations, and a low rate of loss of heterozygosity (LOH) of the tumor suppressors, TP53 and CDKN2A [69]. Equally important, Mueller et al. recently observed that human PanINs (hPanINs) also display allelic imbalances in the KRAS oncogene. More specifically, KRAS mutant allele frequencies higher than 50% were detected in more than half of the examined hPanIN1a and hPanIN2. However, it is estimated that the actual frequency of the mutant copies is higher due to the contamination of healthy tissue in the microdissected region [52]. This proves the presence of amplifications in mutant KRAS already, during the early development of pancreatic cancer.

As the increased allelic frequency of mutant KRAS is also detected in low-grade PanINs and KRAS amplification takes place early in tumorigenesis, it seems that increased mutant KRAS dosage provides a clonal advantage (Figure 4) [52]. However, further changes in KRAS allelic states in PDAC during the evolution of tumors might follow distinct routes that are accompanied by the inactivation of tumor suppressors [70]. It is estimated that the loss of CDKN2A precedes mutant KRAS amplification [51,52]. According to the interplay between macroevolution and punctuated events during PDAC progression, an initial preneoplastic diploid phase precedes genomic instability and copy number alterations. Thus, KRAS point mutations are then followed by large structural rearrangements that are responsible for losses of the wt KRAS allele and other key driver genes. Next, genome doubling is another key molecular event that amplifies existing KRAS gains, leading to major allelic imbalances [51]. These genomic events drive rapid tumor progression and metastasis (Figure 4). Indeed, major KRAS allelic imbalances are detected in a considerable fraction of advanced-stage tumors and hepatic metastases [51,52]. 

Undoubtedly, allelic changes of KRAS are a frequent event in PDAC [71]. Studies involving genetic mouse models have demonstrated their impact in tumor growth and the metastatic capacity of pancreatic cells [51,52,72,73]. Not surprisingly, these functional studies have validated the oncosuppressive role of wt KRAS, as previously suggested [74]. However, the different cellular processes and molecular pathways mediated by the two alleles are still unknown. In other types of cancer that include colorectal, lung, and leukemia, KRAS allelic imbalances are also linked to tumor initiation, cell proliferation, invasion, and metastasis [72,75,76]. Yet, further studies are needed to explore how both the absence of the wt allele and increased dosage of oncogenic KRAS affect the interaction of tumor cells with the stromal and inflammatory microenvironment. This could shed light on the molecular mechanisms of the high metastatic capacity that characterizes these tumors.

### 3.2. Response to Therapy

#### 3.2.1. Prognostic Value and Standard Chemotherapy

Overall, the tumor-specific allelic ratio and dosage of mutant KRAS influence PDAC biology and disease progression [71]. Indeed, it has been strongly demonstrated that PDAC tumors with genomic gains of KRAS mutant alleles have generally worse survival rates [51,77,78]. However, even if the increased mutant allele dosage is a notable predictor of poor survival, it has not shown strong prognostic value across the disease stages [78]. Similarly, the loss of the wild-type allele, often accompanied with copy number gain of mutant KRAS, does not seem to be representative of the overall survival of patients [78]. Thus, so far, the allelic status of KRAS is not included in the decision-making in terms of the therapeutic options and strategies for either resectable or advanced pancreatic cancer. Moreover, there is no evidence on whether tumors with distinct KRAS allelic frequencies may display a different response to the available cytotoxic chemotherapy. It is also worth mentioning that, among the primary resectable cases, the prevalence of patients that receive neoadjuvant therapy is low [78]. Therefore, KRAS allelic status could be a candidate biomarker for the decision of neo-adjuvant treatment or primary resection. Related studies can open up a line of research on personalized medicine to improve the clinical picture of the disease. 

#### 3.2.2. Targeted Therapies and Oncogenic Signaling 

Lennerz and Stenzinger proposed in 2015 that both the allelic frequencies and genomic dosage of mutated KRAS affect PDAC biology [79]. Later on, transcriptomic analysis showed that PDAC cells with an increased KRAS dosage display differences in the downstream oncogenic signaling cascades of the Mitogen-activated Protein Kinase (MAPK) pathway, axon guidance, and Phosphoinositide 3-Kinase (PI3K), compared to cells with a balanced allelic status (Figure 5) [52]. Equally important, additional evidence has validated the function of wild-type KRAS as a tumor suppressor, antagonizing the mutant allele in KRAS dimerization and other biochemical properties [52,72,80]. In most of these studies, it was clearly demonstrated that the downstream signaling of the MAPK pathway is affected [52,72]. Indeed, pharmacological inhibition of MEK with the use of Trametinib and Selumetinib sensitized tumor growth in PDAC and lung mouse models [72,73]. On the contrary, the restoration of wt KRAS expression in human PDAC cells with LOH significantly attenuated the malignancy of tumor cells, confirming the tumor suppressive role of the wt allele [81]. Yan et al. also identified the HIPPO signaling pathway to be positively regulated by wt KRAS in PDAC cells [81]. However, other relevant genes and signaling cascades are also involved in the tumor-suppressive function of wt KRAS and need to be further investigated (Figure 5). Undoubtedly, the presence of the wt allele in tumors should be considered for the design of targeted therapeutic approaches. KRAS allele-specific therapies are now possible due to the recent development of allele-specific KRAS inhibitors [8]. Thus, further studies are necessary to target the broad KRAS signaling network with different combination approaches.

#### 3.2.3. Targeted Therapies Against KRAS wt Tumors

While activating mutations in KRAS are present in the majority of PDAC cases, a significant subset of patients (approximately 10%) includes tumors with wild-type (wt) KRAS [14]. This subgroup of PDACs that harbor wt KRAS alleles has historically been overlooked in terms of targeted therapies, due to the absence of mutations in the KRAS oncogene. However, recent research has highlighted that wt KRAS PDAC tumors still undergo significant genetic alterations that activate alternative signaling pathways and drive tumor progression [82,83]. Among these genetic alterations, NRG1 fusions have emerged as a key event in the pathogenesis of wt KRAS PDAC. These fusions produce chimeric proteins that activate oncogenic signaling mediated by ERBB receptors and promote tumor cell proliferation and metastasis [84]. This suggests that PDAC patients with wt KRAS and NRG1 fusions can benefit from the inhibition of the ERBB signaling pathway [84]. Therefore, the identification of such fusions through molecular diagnostics can guide personalized treatment strategies, and include the use of ERBB inhibitors alone or in combination [84]. Indeed, a clinical study recently demonstrated the efficacy of Zenocotuzumab in NRG1 fusion–positive solid cancers [85].

Furthermore, other prevalent alterations of wt KRAS tumors include fusions, amplifications, and mutations in Receptor Tyrosine Kinases (RTKs), and genes involved in the MAPK and PI3K pathways, such as BRAF, FGFR, ERBB2, ALK, MET, and NF1 [14,82]. Multiple mutations are also often detected within DNA damage repair genes, as well as regulators of chromatin remodeling and the cell cycle [82]. From a therapeutic perspective, the identification of these frequent oncogenic fusion events together with targetable mutations, among patients with wt KRAS tumors, has presented an important opportunity to improve genome-guided treatment options for this subgroup of PDAC patients [9,86].

### 3.3. Challenges in Experimental Methodology

Despite the impact of KRAS as an oncogene, studies on KRAS allelic imbalances did not appear until very recently [51,52,76,78]. This is strongly attributed to the development of high-throughput sequencing methods, including whole genomic, exomic, and transcriptomic sequencing. Due to the molecular biology of the KRAS oncogene and the commonly detected point mutations on codon 12, it is often challenging to distinguish between the wt and mutant allele for the study of allelic ratios. Various protocols for allele-specific real-time PCR with customized probes have been developed to assess allelic abundancies of KRAS [87,88,89,90]. In regard to this concept, digital droplet allele-specific PCR (ddPCR) has increased the sensitivity of signal detection in samples with low concentrations of DNA substrates [91]. Taking it a step further, multiplex ddPCR with mutant- and wt-specific hydrolysis probes enables the detection and assessment of the fractional abundance of KRAS mutations in a cost-effective manner [92]. 

Next Generation Sequencing methods (NGS) have made it possible to assess KRAS allelic frequencies in many cohorts of pancreatic tumors. However, it should also be taken into account that most of these studies have assessed allelic imbalances using bulk technologies. This underlines that single-cell sequencing methods are needed to further explore intratumoral allelic imbalances [93]. In regard to this concept, technologies such as single-nucleus DNA sequencing have demonstrated that there is a large correlation with the pseudobulk variant allele frequency of KRAS [94]. Still, single-cell genomic sequencing is often unable to separate KRAS mt and wt alleles. Undoubtedly, research on KRAS biology in the context of allelic imbalances is highly challenging and will require further advanced molecular techniques and carefully designed functional studies. 

Another major challenge in the research of pancreatic cancer includes the presence of the dense tumor microenvironment [95]. The two most distinguishing features are the robust desmoplasia of tumors, and the non-malignant components that constitute almost 80% of the tumor mass [96]. Briefly, the tumor microenvironment of PDAC is characterized by peritumoral desmoplastic stroma, which consists of a complex array of cellular components surrounded by extracellular matrix (collagens, hyaluronic acid, integrins, proteoglycans, and glycoproteins) [97]. The predominant cell types are cancer-associated fibroblasts, tumor-associated macrophages, regulatory T cells, and other cytotoxic, infiltrating immune cells, as well as endothelial cells [98]. This extensive microenvironment makes the study of the genomic characteristics of tumor epithelial cells very challenging. Common preanalytical approaches to exclusively isolate tumor cells include Laser Capture Microdissection (LCM) and cell-separating technologies (FACS, MACS) [99,100,101]. Other post-sequencing tumor cell enrichment methods include the development of bioinformatic tools that enable to assess the tumor cellularity of samples and, more importantly, to distinguish normal cell types in the tumor microenvironment from malignant cells [102,103,104].

## 4. Conclusions

At the time of diagnosis, only 20% of PDACs are localized, while more than 80% are detected metastasized to the liver and lung or spread to regional lymph nodes. Yet, most research on pancreatic cancer has focused on primary and early-stage resectable tumors, given the availability of these samples. Even so, genomic studies have revealed high mutational conservation between PDAC primary and metastatic tumors, with simple somatic variants to be closely concordant in untreated and advanced-disease patients. Thus, understanding the progression from primary PDACs to metastasis requires the extensive study of intratumoral heterogeneity with single-cell analyses [105]. 

In most cancers, complex genomic rearrangements that alter large genomic regions occur frequently during tumor evolution [47,49]. These genomic events, such as chromothripsis, chromoplexy, and genome doubling, give rise to copy number changes and genetically unstable tumors, defined by a large number of structural variants [13,51]. As Connon and Gallinger discuss, the PDAC evolutionary models and timing of progression can be gradual and stepwise or rapid and punctuated [47]. However, structural mutations and copy number alterations in the punctuated progression model can be difficult to interpret due to the large number of genes that are amplified or lost during these regional events [47]. Indeed, the simultaneous loss of genes, such as CDKN2A or SMAD4, and copy number gains of the transcription factor GATA6, or the KRAS oncogene, are frequently detected [51]. How these simultaneous gains and losses of multiple driver genes affect disease progression, organ-specific dissemination, as well as the response to standard chemotherapy, remains to be studied. 

Various recent studies have shown that changes in KRAS allelic frequencies are important both for tumor progression, as well as major biological and clinical PDAC features [51,52,73,78]. Yet, there is still a long way to go. We and others have observed intratumoral gradients in KRAS allelic frequencies in a mosaic fashion (Figure 4) [51]. Thus, as it has been previously explained, more comprehensive analyses of KRAS copy number changes are needed in pure tumor cell populations and single-cell studies [53]. Considering the possible outgrowth of more aggressive minor clones, further studies are also necessary to uncover how subclones with major KRAS allelic imbalances obtain invasive capacity to metastasize. These observations increase the need for optimized KRAS molecular analyses and functional approaches, in order to study the antagonizing properties of mutant and wild-type KRAS alleles in regard to the KRAS signaling network. 

Thus far, treatment with standard cytotoxic drugs is largely ineffective, especially in locally advanced and metastatic disease, highlighting the need for targeted therapies [3]. Currently, KRAS allelic imbalances are not taken into consideration for the design of therapeutic strategies, even though they alter tumor biology. Therefore, it is necessary to study how they define the response not only to existing chemotherapy, but also to new targeted approaches. Stratifying PDACs according to the KRAS allelic status can help define therapeutic vulnerabilities in groups of tumors with different allelic frequencies. Undoubtedly, PDAC is one of the malignant diseases most urgently in need of more personalized treatment, as well as the development of predictive signatures [106]. In regard to this concept of personalized therapy, the KRAS allelic status should also be included in the genetic characterization of tumors, as it may impact prognosis and the therapeutic sensitivity of patients.

## Figures and Tables

**Figure 1 cancers-17-01226-f001:**
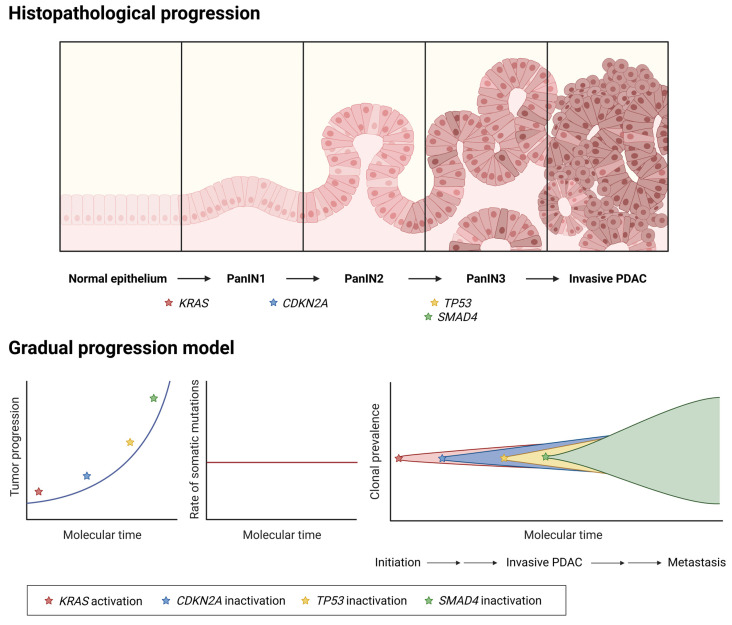
Schematic representation of the gradual progression model of PDAC evolution. Histopathological progression, accumulation rate of somatic mutations, and clonal prevalence over molecular time, are represented.

**Figure 2 cancers-17-01226-f002:**
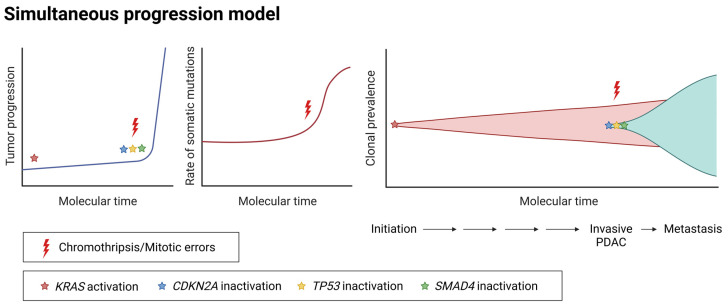
Schematic representation of the simultaneous progression model of PDAC evolution. Tumor progression, accumulation rate of somatic mutations, and clonal prevalence over molecular time, are represented.

**Figure 3 cancers-17-01226-f003:**
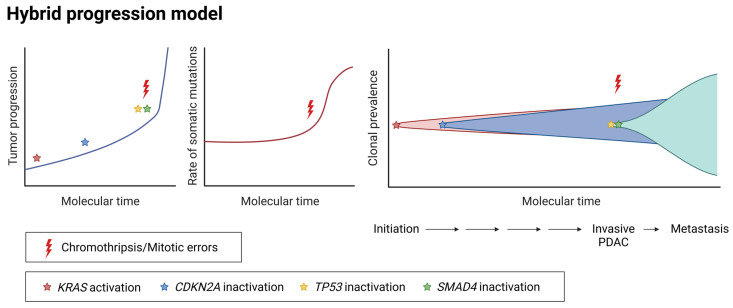
Schematic representation of a hybrid model of PDAC evolution, which combines features of the gradual and simultaneous progression process. Tumor progression, accumulation rate of somatic mutations, and clonal prevalence over molecular time, are represented. Adapted from [47].

**Figure 4 cancers-17-01226-f004:**
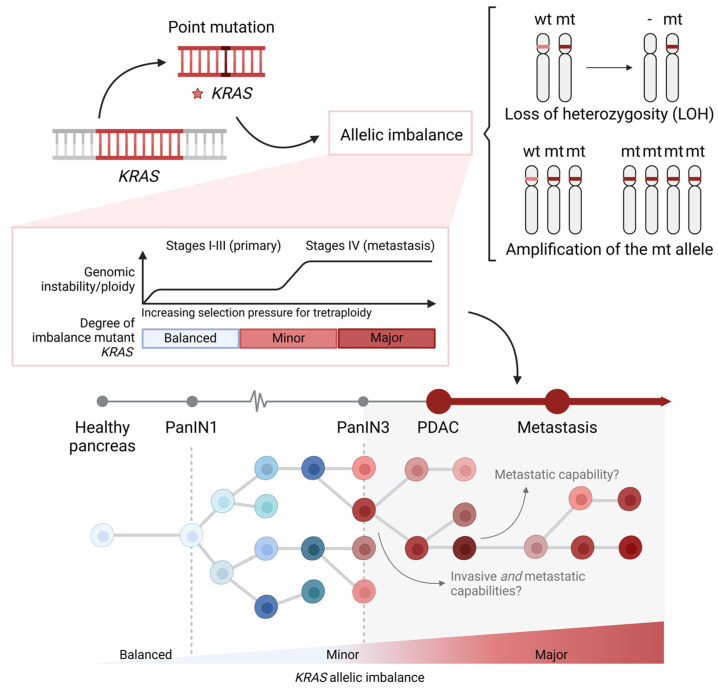
Schematic representation of KRAS allelic imbalances and clonal evolution during tumor progression, from precursor lesions until PDAC and metastasis.

**Figure 5 cancers-17-01226-f005:**
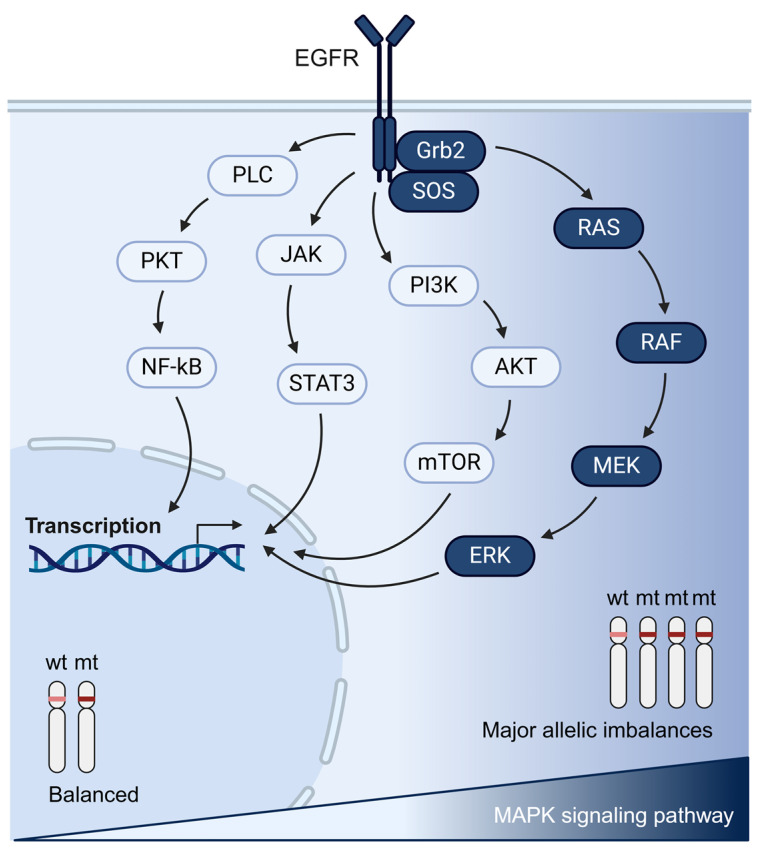
Graphical representation of the RAS signaling network. Increased MAPK signaling is represented in darker blue and is associated with major allelic imbalances and the increased dosage of oncogenic KRAS.

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
