# Peer review of "Unlocking the Genetic Secrets of Pancreatic Cancer: *KRAS* Allelic Imbalances in Tumor Evolution"

_cancers, 2025, doi:10.3390/cancers17071226_

Round 1

Reviewer 1 Report

Comments and Suggestions for Authors

It was a pleasure for me to review this very well written manuscript. In the clinical management of pancreatic cancer the authors propose a personalized therapy, KRAS allelic status should also be included in the genetic characterization of tumors, as it may impact prognosis and therapeutic sensitivity.

Minor revision/proposal:

1. What do the authors believe about primary resectable pancreatic cancer? Should KRAS allelic status be included in the decision about neoadjuvant chemotherapy followed by surgical resection or should these patients undergo primary resection. The latter could have a major clinical impact on the decision-making about this collective 

Author Response

We would like to thank Reviewer 1 for their kind comments. We have included in the manuscript our response to the minor proposal.

Minor revision/proposal:

  1. What do the authors believe about primary resectable pancreatic cancer? Should KRAS allelic status be included in the decision about neoadjuvant chemotherapy followed by surgical resection or should these patients undergo primary resection. The latter could have a major clinical impact on the decision-making about this collective

Response:

It has been recently demonstrated that PDAC tumors with genomic gains of KRAS mutant alleles present generally worse survival (Bielski et al., 2018; Chan-Seng-Yue et al., 2020). However, even if the increased mutant-allele dosage is a notable predictor of poor survival, it has not shown strong prognostic effect across disease stages (Varghese et al., 2025). Similarly, the loss of the wild type allele, often accompanied with copy number gain of mutant KRAS, does not seem to be representative of the overall survival of patients (Varghese et al., 2025). Thus, so far the allelic status of KRAS is not included in the decision-making of the therapeutic options and strategies of either resectable or advanced pancreatic cancer. Moreover, it has not been yet explored whether tumors with distinct KRAS allelic frequencies may display different response to first line chemotherapy. It is also worth mentioning that, among primary resectable cases, the prevalence of patients that receive neoadjuvant therapy is low (Varghese et al., 2025). Therefore, KRAS allelic status could be a candidate biomarker to determinate for neoadjuvant treatment or primary resection. Related studies can open a line of research of personalized medicine to improve the clinical picture of the disease.

Reviewer 2 Report

Comments and Suggestions for Authors

The review is rather short and authors should increase both the context and the number of references.

We would like to encourage authors to increase the following aspect to improve the sound of their manuscript.

 Key aspects:

1. Also evaluate cancer progression from acinar-ductal metaplasia (ADM).

2. Add description of molecular biology methodologies to better discriminate K-Ras allelic imbalance.

3. Consider preanalytical technical approach (e.g. Laser Capture Microdissection) to isolate only epithelial cells.

4. Discuss the criticality related to the low percentage of PDAC patients showing Wt K-Ras phenotype, who could be candidates for a specific therapeutic regimen.

5. Add a large number of references. A review should have at least 100 references.

Author Response

We would like to thank the Reviewer 2 for their useful feedback in order to improve the sound and size of our review. We have included our response to all comments, increased the size of the manuscript and the number of cited references.

Comment 1: Also evaluate cancer progression from acinar-ductal metaplasia (ADM).

Response 1:

As previously described, PanINs are generated from acinar cells through a reprogramming process called acinar to ductal metaplasia (ADM). In normal pancreas, ADM is a temporary and reversible process. However, persistent ADM can lead to PanIN development and eventually PDAC (Marstrand-Daucé et al., 2023). The transition from ADM to PanINs has been well-documented in both mouse models (Hingorani et al., 2003; Guerra et al., 2003; Habbe et al., 2008) and humans (Hruban et al., 2001; Hruban et al., 2007). It is characterized by induction of ductal cell identity and suppression of acinar cell characteristics, both at gene expression and morphology levels (Kopp et al., 2012; Wei et al., 2016). The earliest genetic event that initiates alterations is the oncogenic activation of KRAS, that forces acinar cells into a constant ADM state, which can quickly lead to the formation of PanIN and accelerate PDAC development (Morris et al., 2010; De La O et al., 2009).

Interestingly, these KRAS mutations are detected in healthy pancreatic tissue, further supporting its role as the initiating oncogenic event (Carpenter et al., 2023). Carpenter et al. also demonstrated that ADM and PanINs are surrounded by unique microenvironment. Through spatial transcriptomics, they validated that PanINs are transcriptionally distinct from ducts and are markedly different from ADM (Carpenter et al., 2023). Interestingly, it was previously shown that ADM and PanINs display some of the typical genetic alterations of PDAC. These include aneuploidy, heterozygous mutations and low rate of loss of heterozygosity (LOH) of the tumor suppressors p53 and p16 (Baumgart et al., 2010). Equally importantly, Mueller et al. recently observed that human PanINs (hPanINs) also display allelic imbalances in the KRAS oncogene. More specifically, KRAS mutant allele frequencies higher than 50% were detected in more than half of the hPanIN1a and hPanIN2 examined. However, it is estimated that the actual frequency of the mutant copies is higher due to the contamination of healthy tissue in the microdissected hPanINs (Mueller et al., 2018). This proves the presence of amplifications in mutant KRAS already during the early development of pancreatic cancer.

Comment 2: Add description of molecular biology methodologies to better discriminate K-Ras allelic imbalance.

Response 2: 

Despite the impact of KRAS as an oncogene, studies in KRAS allelic imbalances did not appear until very recently (Burgess et al., 2017; Mueller et al., 2018; Chan-Seng-Yue et al., 2020; Vargese et al., 2025). This is highly attributed to the development of high-throughput sequencing genomic methods, including whole genomic, exomic and transcriptomic sequencing. Due to the molecular biology of the KRAS oncogene and the commonly detected point mutation on codon 12, it is often challenging to distinguish the wt and mutant allele for the study of allelic ratios. Various protocols of allele-specific real-time PCR with customized probes have been developed to assess allelic abundancies of KRAS (Horikoshi et al., 1994; Modrek et al., 2009; Lang et al., 2011; Chubarov et al., 2020). In this concept, digital droplet allele-specific PCR (ddPCR) increased the sensitivity of signal detection in low concentrations of DNA substrates (Denis et al., 2016). Taking it a step further, multiplex ddPCR with mutant- and wt-specific hydrolysis probes enables the detection and fractional abundance of KRAS mutations in a cost-effective manner (Alcaide et al., 2019).

Next generation sequencing methods (NGS) have made it possible to assess KRAS allelic frequencies in many cohorts of pancreatic tumors. However, it should also be taken into account that most of these studies have assessed allelic imbalances with bulk technologies. This underlines that single-cell sequencing methods are needed to further explore intratumoral allelic imbalances. In this concept, technologies such as single-nucleus DNA sequencing has demonstrated to largely correlate with pseudobulk variant allele frequency of KRAS (Zhang et al., 2023). On the other hand, single-cell genomic sequencing is often unable to separate KRAS mt and wt alleles. Undoubtedly, research of KRAS biology in the context of allelic imbalances is highly challenging and will further require advanced molecular techniques and carefully designed functional studies.

Comment 3: Consider preanalytical technical approach (e.g. Laser Capture Microdissection) to isolate only epithelial cells.

Response 3: 

One of the major challenges in the research of pancreatic cancer, includes the presence of the dense tumor microenvironment. The two most distinguishing feautures are the robust desmoplasia of tumors, and the non-malignant components that constitute almost 80% of the tumour mass (Neesse et al., 2011). Briefly, the tumor microenvironment of PDAC is characterized by peritumoral desmoplastic stroma, that consists of a complex array of cellular components surrounded by extracellular matrix (collagens, hyaluronic acid, integrins, proteoglycans and glycoproteins) (Liot et al., 2021). Predominant cell types are cancer-associated fibroblasts, tumor-associated macrophages, regulatory T cells and other cytotoxic, infiltrating immune cells, as well as endothelial cells (Truong and Paulkin, 2021). This extensive microenvironment makes the study of genomic characteristics of tumor epithelial cells very challenging. Common preanalytical approaches to isolate exclusively tumor cells include Laser Capture Microdissection (LCM) and cell separating technologies (FACS, MACS) (Fend and Raffeld, 2000; Boyd et al., 2009; Burdziak et al., 2023). Other post-sequencing tumor-cell-enrichment methods include the development of bioinformatic tools that enable to assess tumor cellularity of samples and more importantly, to distinguish normal cell types of the tumor microenvironment from malignant cells (Cibulskis et al., 2013; Puleo et al., 2017; Gao et al., 2021).

Comment 4: Discuss the criticality related to the low percentage of PDAC patients showing Wt K-Ras phenotype, who could be candidates for a specific therapeutic regimen.

Response 4:

While activating mutations in KRAS are present in the majority of PDAC cases, a significant subset of patients (approximately 10%) harbor tumors with wildtype (wt) KRAS (Raphael et al., 2017). This subgroup of PDACs that harbor wt KRAS alleles has historically been overlooked in terms of targeted therapies, due to the absence of mutations in the KRAS oncogene. However, recent research has highlighted that wt KRAS PDAC tumors still undergo significant genetic alterations that activate alternative signaling pathways and drive tumor progression (Phillip et al., 2022; Topham et al., 2022). Among these genetic alterations, NRG1 fusions have emerged as a key event in the pathogenesis of wt KRAS PDAC. These fusions produce chimeric proteins that activate oncogenic signaling mediated by ERBB receptors, and promoting tumor cell proliferation and metastasis (Heining et al., 2018). This suggests that PDAC patients with wt KRAS and NRG1 fusions can benefit from inhibition of the ERBB signaling pathway (Heining et al., 2018). Therefore, identification of such fusions through molecular diagnostics can guide personalized treatment strategies and include the use of ERBB inhibitors alone or in combination (Heining et al., 2018). Indeed, a clinical study recently demonstrated the efficacy of Zenocotuzumab in NRG1-fusion positive solid cancers (Shram et al., 2025).

Furthermore, other prevalent alterations of wt KRAS tumors include fusions, amplifications and mutations in Receptor Tyrosine Kinases (RTKs), and genes of the MAPK and PI3K pathways, such as BRAF, FGFR, ERBB2, ALK, MET and NF1 (Raphael et al., 2017; Phillip et al., 2022). Multiple mutations are also often detected within DNA-damage repair genes as well as regulators of chromatin remodeling and cell cycle (Phillip et al., 2022). From a therapeutic perspective, the identification of these frequent oncogenic fusion events together with the targetable mutations among patients with wt KRAS tumors, has presented an important opportunity to improve genome-guided treatment options for this subgroup of PDAC patients (Hu et al., 2021).

Comment 5: Add a large number of references. A review should have at least 100 references.

Response 5: We have updated the list of references and citations in the revised manuscript.